# Investigation of oral macrolide prescriptions in Japan using a retrospective claims database, 2013–2018

**Satoshi Ide**[1,2]*, **Masahiro Ishikane**[3,4], **Kensuke Aoyagi**[4], **Akane Ono**[4], **Yusuke Asai**[4], **Shinya Tsuzuki**[4], **Yoshiki Kusama**[5], **Yoshiaki Gu**[6], **Eiichi Kodama**[7], **Norio Ohmagari**[1,3,4]

1 Department of Emerging and Reemerging Infectious Diseases, Graduate School of Medicine, Tohoku University, Sendai, Japan, 2 Division of Infection Control and Prevention, Tokyo Medical and Dental University Hospital, Bunkyo City, Japan, 3 Disease Control and Prevention Center, National Center for Global Health and Medicine, Shinjuku City, Tokyo, Japan, 4 AMR Clinical Reference Center, Disease Control and Prevention Center, National Center for Global Health and Medicine, Shinjuku City, Tokyo, Japan, 5 Department of Pediatric General Medicine, Hyogo Prefectural Amagasaki General Medical Center, Amagasaki, Hyogo, Japan, 6 Department of Infectious Diseases, Graduate School of Medical and Dental Sciences, Tokyo Medical and Dental University, Bunkyo City, Tokyo, Japan, 7 Department of Infectious Diseases, International Research Institute of Disaster Science, Graduate School of Medicine, Tohoku Medical Megabank Organization, Tohoku University, Sendai, Japan

* babytiger0313@gmail.com

**Data Availability Statement:** The data is available from Open Science Framework. DOI: 10.17605/OSF.IO/GK98S.

**Funding:** The authors received no specific funding for this work.

## Abstract

Macrolide usage in Japan exceeds that in Europe and the United States. Investigating the actual conditions in which macrolides are used is important for identifying further interventions for appropriate antimicrobial use; however, this situation has not been evaluated in Japan. Therefore, we aimed to clarify the number of macrolide prescriptions and their changes before and after implementation of the Antimicrobial Resistance (AMR) Action Plan. In addition, we also investigated the names of diseases for which macrolides have been prescribed and the number of days of prescription. A retrospective observational study was conducted using JMDC claims data from January 2013 to December 2018. The proportion of all oral antimicrobials and macrolides used during this period and the diseases for which macrolides were used in the 3 years before and after the AMR Action Plan were determined separately for acute (< 14 prescription days) and chronic (> 14 prescription days) diseases. The number of prescriptions for macrolides constituted approximately 30% of those for all oral antimicrobials; of these, clarithromycin accounted for approximately 60%. Most prescriptions for acute diseases were for common cold, whereas allergic and dermatological diseases were included among chronic diseases. The names of these illnesses did not change before and after the AMR Action Plan. Overall, these results indicate that appropriate macrolide use involves a review of their use for common cold along with appropriate evaluation of their long-term use for skin and allergic diseases. They also indicate the need for further fact-finding studies and ongoing AMR measures.

**Competing interests:** The authors have declared that no competing interests exist.

## Introduction

Since the first antibiotic, penicillin, was introduced in the 1940s, antimicrobials have been used to treat many patients with infectious diseases. Around the year 2000, antimicrobial resistance (AMR), by which existing antimicrobials are rendered ineffective owing to increased and inappropriate use, began to increase [1]. Today, AMR is a major global threat and contributor to disease burden in medical practice [2,3]. By 2050, if no action is taken against AMR, it is expected to cause 10 million deaths yearly worldwide, which exceeds the number of deaths caused by cancer [4,5].

The World Health Organization (WHO) issued a warning of "Antimicrobial Resistance: No Action Today, No Cure Tomorrow" on World Health Day 2011 in response to the global crisis regarding the increase and spread of AMR [6]. Furthermore, at the World Health Assembly in May 2015, a Global Action Plan on Drug Resistance (AMR) was adopted, which called on member countries to develop their own action plans within 2 years [7]. In response, the Japanese government announced the National Action Plan on AMR (hereafter called the Action Plan) in 2016 [8]. This Action Plan set goals related to a total of six areas: public awareness and education, surveillance and monitoring, infection prevention and control, appropriate use of antimicrobials, research and development, and international cooperation. Furthermore, as outcome indicators, numerical targets were set for reducing the use of antimicrobials in humans and reducing AMR in major microorganisms [8]. As inappropriate use of antimicrobials is one of the main factors contributing to AMR emergence, promotion of the Antimicrobial Stewardship Program (ASP) has also been identified as a major strategy [7,9].

Surveillance data on antimicrobial use (AMU) in Japan from 2009 to 2013 showed that oral AMU accounted for 92.6% of the total, and that 77.1% of oral AMU involved broad-spectrum antimicrobials, including third-generation cephalosporins, macrolides, and fluoroquinolones [10]. In 2016, macrolides (32%) were the most used antimicrobials over cephalosporins (28%) in Japan [11].

The defined daily dose (DDD) is the assumed average maintenance dose per day for a drug used for its main indication in adults [12]. The DDD/1000 inhabitants/day (DID) is a measure of the number of people and is used to correct for differences in the usage of different antimicrobials in different populations [12]. In 2013, the DID for macrolides in Japan was 4.83, whereas that in the European Union (EU) and European Economic Area (EEA) was 3.1, indicating higher use [13,14]. The Action Plan aimed to reduce the use of oral cephalosporins, fluoroquinolones, and macrolides by 50% in 2020 compared with that in 2013; however, this target was not achieved, as only 39.5% reduction in macrolide use was found in 2020 [8,15].

Overall, macrolides are an important target for the ASP and should continue to be addressed in the future. However, to date, the situation in which macrolides are prescribed in Japan, including the diseases for which macrolides are used, remain to be surveyed. Therefore, the purpose of this study was to investigate the number and breakdown of macrolide prescriptions and their changes before and after the Action Plan. Along with short-term use to treat acute diseases such as acute upper respiratory tract infections, macrolides are sometimes used for the long-term treatment of chronic diseases, such as diffuse bronchitis and asthma. Therefore, we also investigated the number of days of prescription. For evaluation, we used the JMDC database, a fully anonymized, nationwide electronic database based on receipts from multiple health insurance companies, which covers approximately 2% of the Japanese population and mainly enrolls people in the age range of 0 to 75 years.

## Materials and methods

### Ethics

As this was a database study that did not deal with the personal information of patients, ethical approval was waived.

### Study design and data source

This retrospective observational study used administrative claims data from JMDC Corporation (Tokyo, Japan), covering January 2013 to December 2018. The JMDC database is a fully anonymized, nationwide electronic database based on receipts (inpatient, outpatient, and prescription) from multiple health insurance companies. The database covers approximately 2% of the Japanese population and 10% of health insurance group subscribers, with a cumulative total of 14 million subscribers as of April 2022 [16,17]. The age of enrolled people ranges from 0 to 75 years, but the greatest number of people are aged 15–65 years and belonged to the working population. The data are divided into multiple data sets, including patient information (patient ID, date of birth, gender), facility information (medical facility ID, number of beds code, department code, management entity code), receipt (patient ID, year and month of treatment, medical facility ID, department code), injury and disease information (patient ID, year and month of treatment, medical facility ID, International Classification of Diseases, 10th Revision [ICD-10] code), drug information (patient ID, medical treatment date, medical facility ID, Anatomical Therapeutic Chemical Classification System [ATC] code, dosage and number of days per prescription, route of administration), and prescription status (inpatient or outpatient). As antimicrobials prescribed in Japan mainly include oral antimicrobials and a large proportion of these prescriptions are for outpatient use [10], only the codes for outpatient prescriptions and oral antimicrobials were extracted in this study. Systematic classification of antimicrobials and the names of their components were based on the ATC system (https://www.whocc.no/atc_ddd_index/) of the WHO.

### Data processing

The JMDC dataset was searched by patient ID or medical facility ID, and those with systemic antimicrobial (ATC: J01) use were extracted. Furthermore, a database was created for macrolide antimicrobials (ATC: J01FA) alone. Seven different oral macrolide agents were used during the study period, including erythromycin (J01FA01), spiramycin (J01FA02), roxithromycin (J01FA06), josamycin (J01FA07), clarithromycin (J01FA09), azithromycin (J01FA10), and rokitamycin (J01FA12). Of these, josamycin, spiramycin, and rokitamycin were denoted as 'other macrolides' because their use was negligible.

To investigate the use and changes in macrolides before and after the Action Plan, the types of macrolides and names of diseases (e.g., International Statistical Classification of Diseases, 10th revision [2009]) were analyzed. As the population in the JMDC database is increasing annually and it is difficult to compare annual usage by number of cases, we used proportions for comparison, taking the number of annual prescriptions of macrolides as the denominator [17,18].

### Definition of variables

In Japan, the National Health Insurance covers almost the entire population; furthermore, local governments subsidize pediatric healthcare. To consider the potential difference in antibiotic prescription rates induced by health subsidies from local governments, patients were categorized into 10 age groups (0–3, 4–6, 7–12, 13–18, 19–29, 30–39, 40–49, 50–59, 60–69, and 70–75 years of age) and the proportion of macrolide prescription was evaluated [17].

The macrolides were effective against most gram-positive cocci, some gram-negative rods (*Bordetella pertussis*, *Helicobacter pylori*, *Campylobacter jejuni*, etc.), atypical pathogens (*Mycoplasma pneumoniae*, *Legionella pneumophila*, *Chlamydophila pneumoniae*, *Babesia microti*, *Ureaplasma* spp., etc.), and non-tuberculous mycobacteria [19–21]. The required treatment duration of the antimicrobial therapy varied greatly depending on the disease, severity, immunodeficiency, and other background conditions; however, the optimal duration of therapy has not been determined for some diseases. In severe *M. pneumoniae* pneumonia and *L. pneumophilia* pneumonia in immunocompromised individuals, macrolide usage is recommended for up to 14 days [22,23]. Therefore, in this study, we defined acute diseases as those for which the number of prescribed days was less than 14 and chronic diseases as those for which the number of prescribed days was more than 14 days.

## Statistical analysis

First, we determined the proportion of prescriptions for all antimicrobials and for macrolides from 2013–2018. Next, basic information regarding whether macrolides were prescribed in the 3 years before and after the Action Plan (2013–2015 and 2016–2018) was obtained. Finally, we identified the names of diseases for which macrolides were prescribed in the 3 years before and after the Action Plan, separately for acute and chronic diseases.

The median and interquartile range (IQR) were calculated for continuous variables, and percentages (%) were determined for categorical variables. For Figs 1B and 2, the Cochran-Armitage trend test was performed. All analyses were performed using the Stata/MP 17.0 software (StataCorp LLC, College Station, TX, USA).

As the objective of this study was to determine the epidemiology of macrolide prescriptions, only descriptive epidemiology was performed without statistical evaluation, because the sample size in this study was large and could easily reach statistical significance.

## Results

### Changes in proportions of antimicrobials prescribed from 2013 to 2018

Fig 1A shows the number of prescriptions for all antimicrobials prescribed from 2013 to 2018, and Fig 1B shows the proportion of the type of prescribed antimicrobials. The numbers of

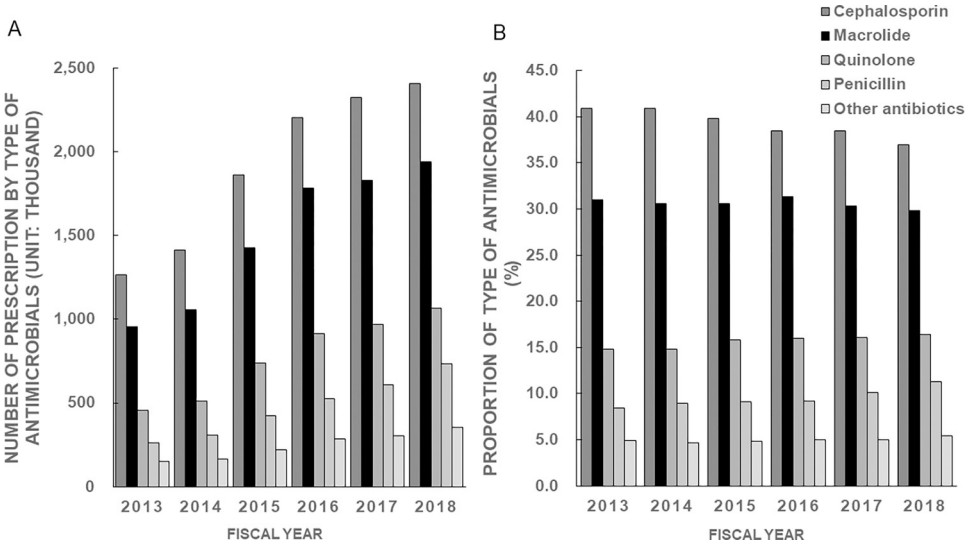

**Fig 1. Trend and type of all prescribed antimicrobials during 2013–2018.** (A) Number of prescriptions for different types of antimicrobials. (B) Proportions of different types of antimicrobials.

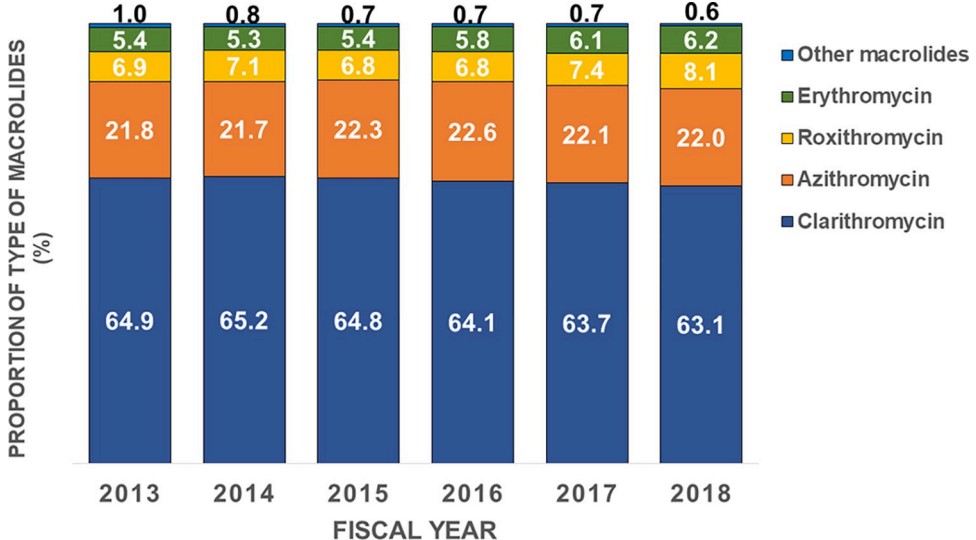

**Fig 2. Trend and type of all prescribed macrolides during 2013–2018.**

prescriptions for macrolides nearly doubled from 958,028 in 2013 to 1,939,474 in 2018. The proportion of macrolide prescriptions was approximately 30% from 2013 to 2018. The proportion of prescriptions for cephalosporins and macrolides was significantly decreased (p<0.001), whereas that for quinolones and penicillins was significantly increased (p<0.001). The proportions of types of prescribed macrolides by year from 2013 to 2018 are also shown in Fig 2. Clarithromycin accounted for the largest proportion (more than 60%) in all years. The proportion of prescriptions for clarithromycin was significantly decreased (p<0.001), whereas that for azithromycin, roxithromycin, and erythromycin was significantly increased (p<0.001).

## Changes in the proportion of macrolide prescriptions by age from 2013 to 2018

The number of prescriptions for macrolides from 2013 to 2018, broken down by age group, is shown in Fig 3. Prescriptions for clarithromycin accounted for approximately 60% of the total and were among the highest in all age groups.

In age groups of 0–3 and 19–29 years, in which clarithromycin was prescribed at less than 60%, relatively higher prescriptions of erythromycin (19.4%) and roxithromycin (14.3%) were found. Azithromycin was prescribed relatively frequently in the age group 19–29 and 30–39 years, and exceeded 25%.

## Basic information on macrolide prescriptions in the 3 years before and after the Action Plan

The characteristics of the patients and facilities at which macrolides were prescribed during the entire study period (n = 13,657,028), 2013–2015 (n = 5,242,369), and 2016–2018 (n = 8,414,659) are shown in Table 1. Overall, 6,505,157 (47.4%) of patients were male, with a median (IQR) age of 39 (16–54) years. Dividing the patients into 10 groups based on age, the groups in their 30s and older (excluding the 70–75 year age group) accounted for more than 10%, with the 50–59 years group accounting for the largest number of cases (2,276,643; 17.1%). Internal medicine was the most common department to prescribe (6,889,473 cases; 50.4%), followed by otolaryngology with 1,216,313 cases (8.9%), and pediatrics with 907,866

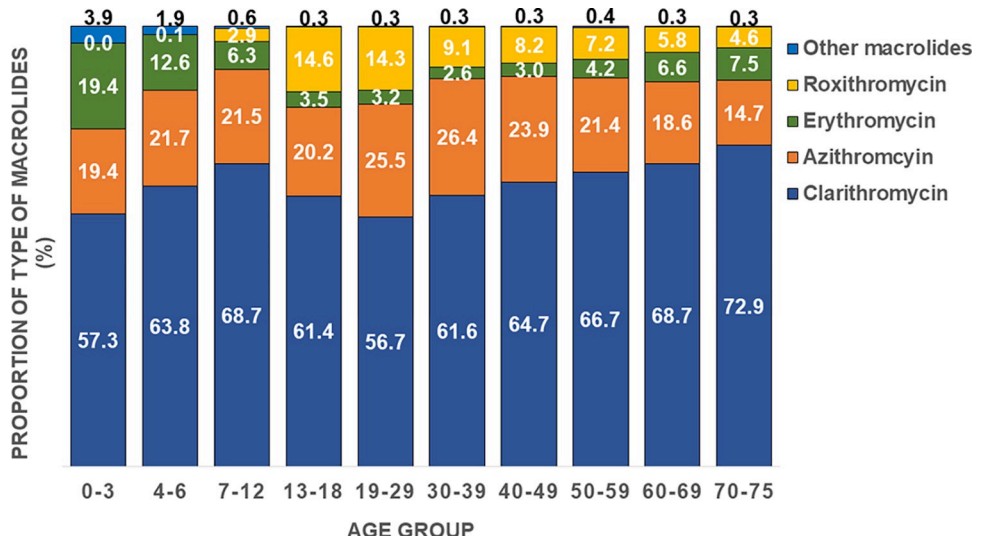

**Fig 3. Proportion of prescribed macrolides among all macrolides as a function of age during 2013–2018.**

**Table 1. Characteristics of patients and facilities at which macrolides were prescribed during 2013–2018.**

| Characteristics | | Total number of macrolide prescriptions (n = 13,657,028) | 2013–2015 (n = 5242369) | 2016–2018 (n = 8414659) |
|---|---|---|---|---|
| **Patient characteristics** | | | | |
| Age, median | | 39 (IQR: 16–54) | 37 (IQR: 14–53) | 40 (IQR: 17–54) |
| Male sex | | 6,505,157 (47.6%) | 2,509,171 (47.8) | 3,901,476 (46.4%) |
| Age group (years) | | | | |
| 0–3 | | 971,371 (7.7%) | 426,898 (8.1) | 544,473 (6.5%) |
| 4–6 | | 804,357 (6.4%) | 346,804 (6.6) | 457,553 (5.4%) |
| 7–12 | | 1,129,333 (8.3%) | 454,610 (8.7) | 674,723 (8.0%) |
| 13–18 | | 876,512 (6.4%) | 334,263 (6.4) | 542,249 (6.4%) |
| 19–29 | | 1,341,620 (9.8%) | 503,846 (9.6) | 837,774 (10.0%) |
| 30–39 | | 1,849,855 (13.5%) | 730,151 (13.9) | 1,119,704 (13.3%) |
| 40–49 | | 2,276,643 (16.7%) | 851,294 (16.2) | 1,425,349 (16.9%) |
| 50–59 | | 2,339,225 (17.1%) | 843,566 (16.1) | 1,495,659 (17.8%) |
| 60–69 | | 1,646,147 (12.0%) | 588,099 (11.2) | 1,058,048 (12.6%) |
| 70–75 | | 421,965 (3.3%) | 162,838 (3.1) | 259,127 (3.1%) |
| **Facility characteristics** | | | | |
| Specialty | | | | |
| | Internal medicine | 6,889,473 (50.4%) | 2,642,415 (50.4%) | 4,247,058 (50.5%) |
| | Otolaryngology | 1,216,312 (8.9%) | 497,391 (9.5%) | 718,921 (8.5%) |
| | Pediatrics | 907,866 (6.6%) | 365,673 (7.0%) | 542,193 (6.4%) |
| | Others | 4,643,377 (34.0%) | 1,745,621 (33.3%) | 2,897,756 (34.4%) |
| Operation type | | | | |
| | Clinic | 8,963,226 (65.6%) | 3,513,813 (67.0%) | 5,449,413 (64.8%) |
| | National and public hospital | 1,133,361 (8.3%) | 427,111 (8.1%) | 706,250 (8.4%) |
| | University hospital | 951,141 (7.0%) | 317,454 (6.1%) | 633,687 (7.5%) |
| | Other hospital | 2,608,793 (19.1%) | 983,991 (18.8%) | 1,624,802 (19.3%) |
| | No data | 507 (0%) | 0 (0%) | 507 (0%) |

cases (6.6%). In terms of facility operation type, clinics accounted for the most prescriptions (8,963,226 cases; 65.6%); however, data were missing for 507 cases.

Comparisons of the proportion of prescriptions between 2013–2015 and 2016–2018 showed a decrease in the 0–3, 4–6, and 7–12 years age groups from 8.1%, 6.6%, and 8.7% to 6.5%, 5.4%, and 8.0%, respectively. In contrast, the proportions of prescriptions for patients aged 40–49, 50–59, and 60–69 years increased from 16.2%, 16.1%, and 11.2% to 16.9%, 17.8%, and 12.6%, respectively. The proportion of prescriptions from otolaryngology and pediatrics also decreased from 9.5% and 7.0% to 8.5% and 6.4%, respectively. Regarding the type of medical institution, prescriptions from clinics decreased from 67.0% to 64.8%, whereas those from university hospitals increased from 6.1% to 7.5%.

## Epidemiology of diseases for which macrolides were prescribed in the 3 years before and after the Action Plan

Table 2 shows the 10 most common acute diseases for which macrolides were prescribed in 2013–2015 (n = 13,555,601) and 2016–2018 (n = 23,684,663). Clarithromycin was the most commonly prescribed macrolide for 9 of the 10 diseases on the list. Except for asthma, all were respiratory infections and eight were acute diseases. A comparison of 3 years before and after the Action Plan showed that the disease names were generally consistent, although there were some differences in the percentages and rankings of illnesses.

A similar analysis was performed, and the 10 most common chronic diseases for which macrolides were prescribed in 2013–2015 (n = 1,793,965) and 2016–2018 (n = 3,660,871) [Table 3]. Clarithromycin was also the most commonly prescribed macrolide for 9 of the 10 diseases, and roxithromycin was the most commonly prescribed for 1 disease. In the breakdown of diseases, in 2013–2015, five diseases were respiratory infections (chronic sinusitis, acute sinusitis, acute laryngopharyngitis, acute bronchitis, non-purulent otitis media), two were dermatologic diseases (acne vulgaris, dermatitis), two were allergic disease (allergic rhinitis, asthma), and one was an ophthalmic disease (acute atopic conjunctivitis). In 2016–2018,

**Table 2. List of acute diseases for which macrolides were prescribed for less than 14 days.**

| | 2013–2015 | | | | | 2016–2018 | | | | |
|---|---|---|---|---|---|---|---|---|---|---|
| Rank | Drug | ICD10 | Name of disease | No. of cases | Proportion (%) | Drug | ICD10 | Name of disease | No. of cases | Proportion (%) |
| 1 | CAM | J209 | Acute bronchitis, unspecified | 995,407 | 7.3 | CAM | J304 | Allergic rhinitis, unspecified | 1,657,206 | 7.0 |
| 2 | CAM | J304 | Allergic rhinitis, unspecified | 945,240 | 7.0 | CAM | J209 | Acute bronchitis, unspecified | 1,612,215 | 6.8 |
| 3 | CAM | J459 | Unspecified asthma | 564,155 | 4.2 | CAM | J459 | Unspecified asthma | 917,233 | 3.9 |
| 4 | CAM | J060 | Acute laryngopharyngitis | 459,862 | 3.4 | CAM | J060 | Acute laryngopharyngitis | 806,678 | 3.4 |
| 5 | CAM | J019 | Acute sinusitis, unspecified | 409,885 | 3.0 | CAM | J019 | Acute sinusitis, unspecified | 711,505 | 3.0 |
| 6 | CAM | J029 | Acute pharyngitis, unspecified | 401,783 | 3.0 | CAM | J029 | Acute pharyngitis, unspecified | 704,242 | 3.0 |
| 7 | CAM | J069 | Acute upper respiratory infection, unspecified | 376,362 | 2.8 | CAM | J069 | Acute upper respiratory infection, unspecified | 635,937 | 2.7 |
| 8 | CAM | J329 | Chronic sinusitis, unspecified | 265,432 | 2.0 | CAM | J329 | Chronic sinusitis, unspecified | 448,453 | 1.9 |
| 9 | AZM | J209 | Acute bronchitis, unspecified | 261,358 | 1.9 | AZM | J209 | Acute bronchitis, unspecified | 434,313 | 1.8 |
| 10 | CAM | J00- | Acute nasopharyngitis [common cold] | 215,737 | 1.6% | CAM | J111 | Influenza due to unidentified influenza virus with other respiratory manifestations | 419,588 | 1.8% |

Here, 2013–2015 and 2016–2018 indicate years and refer to January 2013 to December 2015 and January 2016 to December 2018, respectively.

Abbreviations; CAM: Clarithromycin, AZM: Azithromycin.

**Table 3. List of chronic diseases for which macrolides were prescribed for more than 14 days.**

| | 2013–2015 | | | | | 2016–2018 | | | | |
|---|---|---|---|---|---|---|---|---|---|---|
| Rank | Drug | ICD10 | Name of disease | No. of cases | Proportion (%) | Drug | ICD10 | Name of disease | No. of cases | Proportion (%) |
| 1 | CAM | J304 | Allergic rhinitis, unspecified | 139,575 | 7.8 | CAM | J304 | Allergic rhinitis, unspecified | 280,679 | 7.7 |
| 2 | CAM | J329 | Chronic sinusitis, unspecified | 108,738 | 6.1 | CAM | J329 | Chronic sinusitis, unspecified | 222,921 | 6.1 |
| 3 | CAM | J459 | Asthma, unspecified | 53,694 | 3.0 | CAM | J459 | Asthma, unspecified | 108,300 | 3.0 |
| 4 | CAM | J019 | Acute sinusitis, unspecified | 50,977 | 2.8 | CAM | J019 | Acute sinusitis, unspecified | 97,838 | 2.7 |
| 5 | RXM | L700 | Acne vulgaris | 46,752 | 2.6 | CAM | J060 | Acute laryngopharyngitis | 90,070 | 2.5 |
| 6 | CAM | J060 | Acute laryngopharyngitis | 43,554 | 2.4 | RXM | L700 | Acne vulgaris | 84,985 | 2.3 |
| 7 | CAM | J209 | Acute bronchitis, unspecified | 32,961 | 1.8 | CAM | J209 | Acute bronchitis, unspecified | 66,442 | 1.8 |
| 8 | CAM | H101 | Acute atopic conjunctivitis | 22,980 | 1.3 | CAM | H101 | Acute atopic conjunctivitis | 48,376 | 1.3 |
| 9 | CAM | H659 | Nonsuppurative otitis media, unspecified | 21,330 | 1.2 | CAM | J42 | Unspecified chronic bronchitis | 38,737 | 1.1 |
| 10 | CAM | L309 | Dermatitis, unspecified | 19,475 | 1.1 | CAM | H659 | Nonsuppurative otitis media, unspecified | 38,148 | 1.0 |

Here, 2013–2015 and 2016–2018 indicate years and refer to January 2013 to December 2015 and January 2016 to December 2018, respectively.

Abbreviations; CAM: Clarithromycin, RXM: Roxithromycin.

six diseases were respiratory infections (chronic sinusitis, acute sinusitis, acute laryngopharyngitis, acute bronchitis, chronic bronchitis, non-purulent otitis media), two were allergic diseases (allergic rhinitis, asthma), one was a skin disease (acne vulgaris), and one was an ophthalmic disease (acute atopic conjunctivitis). In each period, all the diseases were acute. In all periods, five diseases included "acute" in their name. Sinusitis was the most common acute and chronic condition in both periods.

## Discussion

In this study, we analyzed Japanese social insurance data from the JMDC and clarified the epidemiology of the number of macrolide prescriptions. The number of macrolide prescriptions accounted for approximately 30% of all oral antimicrobial prescriptions; among these, clarithromycin was the most common, similar to previous reports [11,24]. Comparing the situation in other countries, among outpatient prescriptions of antimicrobials in the United States, macrolides account for 23%, and azithromycin is the most prescribed antimicrobials [25]. In Europe, intermediate-acting macrolides, mainly comprising clarithromycin, show the largest prescription rate at 58.9%, suggesting that the situation in Japan and Europe is similar [13].

Our study shows a decrease in the proportion of macrolide prescriptions from 2013 to 2018 (Fig 1B), although the overall number of cases increased each year (Fig 1A). Clarithromycin showed a decreasing trend, while azithromycin, roxithromycin, and erythromycin showed increasing trends (Fig 2). The 10 most common acute diseases for which macrolides were prescribed showed no change before and after the Action Plan. National antimicrobial sales in Japan have been reported to have decreased since 2016 [26]. Therefore, it was speculated that the decrease in overall antimicrobial use of antimicrobials for cold syndromes and other conditions due to the AMR measures may have influenced the decrease in the percentage of macrolide antimicrobials used. The reasons clarithromycin is frequently used in Japan is likely because it is a broad-spectrum antibacterial agent with few side-effects, acts for a long time in the body, and is expected to have an immunomodulating effect. Further investigation is needed to clarify the factors that contributed to the decrease in the use of macrolide antibacterial agents [11].

Current AMR measures include antimicrobial awareness surveys, surveillance of antimicrobial use, and the development of guidance and educational tools around the proper use of antimicrobials for primary care physicians [27]. However, there is still no approach targeting reduction of macrolides. To further rationalize macrolide prescriptions, several different and specific approaches are needed. The first is to reduce the use of macrolides against common cold (acute bronchitis, acute laryngopharyngitis, acute sinusitis, acute pharyngitis, acute upper respiratory tract inflammation, acute bronchiolitis, and acute nasopharyngitis). A previous study defined diseases that required the use of macrolides. Moreover, the study classified allergic rhinitis, asthma, acute bronchitis, influenza, non-purulent otitis media, and viral upper respiratory tract infections as those that did not require them. Therefore, it reported that the proportion of diseases for which macrolides should be used as first-line therapy was only 5% [25]. As shown in Table 2, macrolides are commonly used for common cold; therefore, refraining from using macrolides for treating common cold is one possible approach to reduce unnecessary consumption.

The second approach is to review the use of long-term use macrolide prescriptions. As shown in Table 3, similar to the acute phase, use for common cold accounted for half of most prescribed diseases, but macrolides were also used for dermatologic diseases and allergic diseases. Macrolides act against a wide range of bacteria, and 14- and 15-membered macrolides are also known to exert immunomodulatory effects [28]. After reports of long-term low-dose erythromycin use in patients with diffuse panbronchiolitis in Japan, small-dose long-term macrolide therapy was thought to be effective for other chronic airway diseases [29]. Macrolide therapy has been recommended for chronic sinusitis, bronchial asthma, bronchiectasis, and chronic obstructive pulmonary disease in other articles and guidelines, and has been established as a useful therapeutic option [28,30–32]. Furthermore, explanations of problems associated with long-term administration are limited to descriptions related to monitoring adverse effects and the possibility of drug resistance; but these have not been adequately discussed. The macrolide resistance rate against Streptococci, Mycoplasma, and the *Mycobacterium avium* complex is increasing in many countries [33–40]. Moreover, macrolide resistance affects treatment prognosis, especially in non-tuberculous mycobacteria where macrolides are the first-line or key drugs [21,39,40]. In addition, previous reports have suggested that in patients with bronchiectasis and non-cystic fibrosis, long-term treatment with erythromycin may alter the oropharyngeal microflora and increase the level of antimicrobial resistance [41]. In sub-Saharan Africa, a 4-year period of twice-yearly mass distribution of azithromycin to preschoolers was reported to increase resistance rates to azithromycin as well as non-macrolide antimicrobials [42]. There are concerns that even long-term administration of small doses of macrolides like clarithromycin may pose a similar risk, indicating the need for further studies. To minimize emergence of resistance, the evidence for diseases requiring low-dose long-term administration should be reaffirmed, and physicians should be educated avoid prescribing for prolonged cough only.

The third approach involves evaluation of macrolides for treating acne vulgaris, as shown in Table 3, where the use of roxithromycin for acne vulgaris was found to be among the top 10 most common diseases for which macrolides were prescribed for long-term. As of September 2022, roxithromycin was not marketed in the United States and was only available in some countries, including Australia, France, and Germany. In Japan, roxithromycin is also indicated for dermatologic infections, chronic pyoderma, acne vulgaris, laryngopharyngitis, acute bronchitis, pneumonia, otitis media, and sinusitis. Oral antimicrobial therapy is strongly recommended for inflammatory skin rashes in the 2016 acne vulgaris treatment guidelines of the Japanese Dermatological Association [43]. In contrast, oral retinoids are the first choice of treatment for severe acne vulgaris in the US and EU guidelines; however, these are not

approved in Japan owing to their serious side effects, including teratogenicity [44,45]. In case of regulatory approval of oral retinoids by further clinical trials, use of systemic antimicrobials including macrolides may decrease. Furthermore, azithromycin and erythromycin are recommended in the guidelines for moderate to severe acne vulgaris, which is refractory to local therapy [44]. Although azithromycin has fewer side effects than tetracyclines, macrolides should only be considered when tetracyclines are unavailable. Moreover, erythromycin use should be limited considering the development of resistance. Furthermore, systemic antimicrobials should be administered for as short a time as possible, with re-evaluation every 3–4 months to prevent the acquisition of resistance [44]. Based on our data, the severity of acne vulgaris and the history of tetracycline administration could not be evaluated. These points may serve as potential interventions for the ASP in future.

This study has several limitations. First, although this study analyzed a large dataset of 13.6 million prescription records, the JMDC database is based on the health insurance of employees of large companies and their families; therefore, the number of subjects younger than 15 years of age and older than 65 years of age is relatively small and the results may not be generalizable. Second, because it is a retrospective, administrative claim database, it lacks clinical information such as symptoms, severity, pathogens, drug allergies, laboratory data, and adherence to antimicrobials. Furthermore, it does not verify whether the disease name is consistent with the actual diagnosis. These shortcomings have also been pointed out in previous studies using JMDC data [17,46,47]. We believe that this issue needs to be addressed in future because there is no medical database with sufficiently verified diagnostic and clinical information in Japan. Third, as the database was integrated by linking the dataset, there may not be a single disease registered at the time of prescribing the macrolide antimicrobials. Furthermore, if a drug other than macrolide antibacterial agents is prescribed at the same visit, several disease names are added to the receipt. Therefore, unrelated comorbidities such as diabetes, hypertension, and insomnia are also added to the list of disease names, making it difficult to determine whether macrolide antimicrobials were truly used. We evaluated the 10 most common diseases for which macrolides were prescribed to avoid this problem. Fourth, the results (except for Tables 2 and 3) use the number of prescriptions as the evaluation index and do not consider the number of days of prescription. Thus, even if the total number of prescription days is the same, the number of prescriptions increases if there are multiple prescriptions for a short period, which introduces a difference in the results. Previous studies have also used number of cases; thus, we used the number of cases in the present study for comparison with the past results [17,25].

Despite several limitations, the strengths of this study are that it identifies the epidemiology of macrolide prescriptions and the names of diseases for which macrolides are used by dividing them into acute and chronic based on the number of prescription days. Furthermore, it identifies possible intervention points for appropriate macrolide use in the future. We believe that AMR measures, including further rationalization of macrolide use, are needed to efficiently cure patients, reduce AMR, and reduce harm caused by unnecessary use of antimicrobials [48]. Overall, our results are important in terms of considering the potential of such measures.

In conclusion, macrolides account for approximately 30% of all oral antimicrobials prescribed in Japan, with clarithromycin being the most used. To reduce the use of macrolides, reviewing their use for treating common cold and reevaluating their long-term use for allergic and dermatologic disease may be necessary. These findings are important for appropriate macrolide use in the ASP. However, further research is needed on the actual use of macrolide antimicrobials and the ongoing AMR measures.

## Author Contributions

**Conceptualization:** Satoshi Ide, Masahiro Ishikane, Yoshiki Kusama.

**Data curation:** Satoshi Ide, Kensuke Aoyagi.

**Formal analysis:** Satoshi Ide, Yusuke Asai, Shinya Tsuzuki.

**Investigation:** Satoshi Ide.

**Methodology:** Satoshi Ide, Masahiro Ishikane, Kensuke Aoyagi, Akane Ono, Yoshiki Kusama, Norio Ohmagari.

**Project administration:** Satoshi Ide.

**Supervision:** Eiichi Kodama, Norio Ohmagari.

**Validation:** Masahiro Ishikane.

**Visualization:** Satoshi Ide.

**Writing – original draft:** Satoshi Ide.

**Writing – review & editing:** Masahiro Ishikane, Kensuke Aoyagi, Akane Ono, Yusuke Asai, Shinya Tsuzuki, Yoshiki Kusama, Yoshiaki Gu, Eiichi Kodama, Norio Ohmagari.

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
