## [Decision Letter · Decision Letter 0]

20 Dec 2022

PONE-D-22-29639Investigation of oral macrolide prescriptions in Japan using a retrospective claims database, 2013–2018PLOS ONE

Dear Dr. Ide,

Thank you for submitting your manuscript to PLOS ONE. After careful consideration, we feel that it has merit but does not fully meet PLOS ONE’s publication criteria as it currently stands. Therefore, we invite you to submit a revised version of the manuscript that addresses the points raised during the review process.

Please address all discrepencies and flaws pointed out by the reviewrers. Presentation of data also requires improvement.

We look forward to receiving your revised manuscript.

Kind regards,

Iddya Karunasagar

Academic Editor

PLOS ONE

Journal Requirements:

    "No authors have competing interests"

Additional Editor Comments:

Please see the reviewer comments. The manuscript requires substantial revision in all sections, particularly presentation of results, organisation of manuscript structure and reference citation. Please address the discrepancies between data and presentation in the manuscript. The reviewer comments should be addressed point by point.

Reviewers' comments:

Reviewer's Responses to Questions

**Comments to the Author**

1. Is the manuscript technically sound, and do the data support the conclusions?

Reviewer #1: Yes

Reviewer #2: No

Reviewer #3: Partly

2. Has the statistical analysis been performed appropriately and rigorously? 

Reviewer #1: N/A

Reviewer #2: Yes

Reviewer #3: N/A

3. Have the authors made all data underlying the findings in their manuscript fully available?

Reviewer #1: Yes

Reviewer #2: Yes

Reviewer #3: Yes

4. Is the manuscript presented in an intelligible fashion and written in standard English?

Reviewer #1: Yes

Reviewer #2: No

Reviewer #3: Yes

5. Review Comments to the Author

Reviewer #1: useful study for Japan to implement Antimicrobial stewardship. Trend analysis statements are not clearly supported by statistical analysis. Need to give details of how the trend analysis was performed. Comparing 3 years before and after may not be statistically sound.

Other limitations are mentioned in the manuscript.

Reviewer #2: 2. References are placed beside sentences in a confusing way.

3. The structure of some sentences is contradictory, example paragraph 290- 295.

4. Abbreviations appear without explanation. Their explanation then appear after 2 or three pages; example JMDC in page 3.

5. Explanation of Japan Medical Database Claim (JMDC), the core of the study, should be placed in the introduction; what it is, importance, data contained and comparison with similar databases of other countries. It is not appropriate to be placed in the study design.

6. DDD, and DID

• Reference [12] does not define DID; it defines DDD.

• DID is not a measure of the number of people. Lines 83 and 84 need to be rephrased. DDD and DID should be explained in a clearer way with proper placement of references.

• DDDs are not established for all medicines with an ATC code. Major drug groups without DDDs are, in addition to others, ophthalmologicals and otologicals. Table 1 showing that among characteristics of patients and facilities where macrolides were prescribed are otolaryngology hospitals which are expected to provide treatment for patients with ontological problems. For such patients DDDs are not the right choice. Table 2 shows infections for which macrolides were prescribed and including conjunctivitis.

• Page 10, patients were categorized into 10 age group (0-3, 4-6, 7-12………………) e.g., which are considered children groups. It is to be emphasized that DDD for children is challenging. Assessing frequency of drug use in children is not possible by using DDDs due to the changeability of children’s doses, according to WHO reference https://www.who.int/tools/atc-ddd-toolkit/about-ddd

7. Statistical analysis: statistical analysis of pattern of macrolide prescription which was not done in the current work was necessary in comparison with standard treatment regimens. Such analysis would identify the prescription behaviors of clinicians and could determine rational and rational use of macrolides. It would also show cases where antibiotics should not be prescribed. Such findings would form the objectives for designing educational programs to raise awareness among prescribers, so fulfilling recommendations of the country action plan.

8. Results

In order to add more power to the study, it is recommended to translate your findings into financial cost. This can be done by determining the cost- benefit of eliminating unnecessary use of macrolides. In addition to be clearer, this will help get the message across to prescribing clinicians. These findings could be presented during orientation programs to enhance prudent use of macrolides.

Another important point is to highlight disease where macrolides were to be prescribed, but they were not. It is important to decrease the frequency of macrolides use, and at the same time do not interfere with providing proper healthcare service.

9. Discussion

• What is the reason that clarithromycin was the most commonly prescribed?? Is due to price factor or more activity of drug companies?

• Line 297: what are the current AMR measurses????

• Line 298: is there any rapid test done to exclude bacterial infection in cases of common cold as CRP for example. What do you do if a neutropenic patient is coming with common cold. Clear guidelines should be provided for prescribing macrolides in specific situations.

• As regards long term use, what do you make to minimize emergence of resistance

• Authors listed limitation of the study. The second limitation is a serious one. The question is what is the relation of this study with medicine??? A professional statistician can make similar work. Authors mentioned in lines 361, 362 that their shortcomings were also pointed out in previous studies, so why they did not have different approach. Since there is no data base in Japan with sufficiently verified diagnosis and clinical information, such studies are of no value

• Subsequent limitation starting at line 364, destroys the study design of using DDDs. A finding that is already established since DDD is sometimes a dose that is rarely or never prescribed because it is an average of two or more commonly used doses.

• The fourth limitation (line 71) is again falling in the same mistake of other authors.

• Line 382, a serious concept in providing high quality patient care is the term (decrease prescription), the right term is to (rationalize prescription).

Conclusions:

1. The study could be a part of more meaningful work. It could be the first portion of an interventional study at less wide scale.

Reviewer #3: p9 l135 J01FA12 corresponds to rokitamycin instead of roxithromycin.

Results section contain many errors. The authors quote number of prescirptions as well as proportion and some mix-up occurs. E.g. p12, l187, they mention the number of prescriptions for cephalosporins and penicillins show a decreasing trend ... but in figure 1-A it is definitely increasing!

The section on "Epidemiology of diseases for which macrolides were prescribed in the three years before and after the Action Plan" is not well written. It starts with "Table 2 shows the top 10 most prescribed macrolides for acute illnesses in 2013–2015 (n = 13,555,601) and 2016–2018 (n = 23,684,663). Clarithromycin was prescribed for nine of the 10 top most diseases, and azithromycin was prescribed for one disease. " Firstly, the table shows the 10 most common acute diseases for which macrolides were prescribed and not 10 most prescribed macrolides. Secondly, the authors mean that clarithromycin was THE MOST COMMONLY prescribed macrolide for 9 of the 10 diseases on the list.

Table 2 itself is presented as "List of acute infectious diseases for which macrolides were prescribed for less than 14 days" but includes non-infectious diseases such as asthma and allergic rhinitis.

p14, l229 "Comparisons between 2013–2015 and 2016–2018 showed a decrease in the age groups 0–3, 4–6, and 7–12 years from 8.1%, 6.6%, and 8.7% to 6.5%, 5.4%, and 8.0%, respectively." It is the PROPORTION OF PRESCRIPTIONS which occurred in these age groups which decreased.

In the discussion section, lines 290-292, "Our study revealed a downward trend in macrolide prescription among all

antimicrobials from 2013–2018,..." but the number actually doubled as seen in figure 1A.

Overall, the Introduction, Materials and methods and Discussion sections are acceptable, except for the above comments. Minor revisions are required.

The Results section is not well written and contain inaccuracies, some of which are highlighted above. A major revision of this section is required.

6. PLOS authors have the option to publish the peer review history of their article (what does this mean?). If published, this will include your full peer review and any attached files.

Reviewer #1: **Yes: **Dr Kushlani Jayatilleke

Reviewer #2: No

Reviewer #3: No

---

## [Author Response · Author response to Decision Letter 0]

2 Mar 2023

Please see attached file "response_to_reviewer_comments_PlosOne" for responce to reviewers and editor.

---

## [Decision Letter · Decision Letter 1]

5 May 2023

PONE-D-22-29639R1Investigation of oral macrolide prescriptions in Japan using a retrospective claims database, 2013–2018PLOS ONE

Dear Dr. Ide,

Thank you for submitting your manuscript to PLOS ONE. After careful consideration, we feel that it has merit but does not fully meet PLOS ONE’s publication criteria as it currently stands. Therefore, we invite you to submit a revised version of the manuscript that addresses the points raised during the review process.

Some minor changes needed as recommended by the reviewers 

We look forward to receiving your revised manuscript.

Kind regards,

Iddya Karunasagar

Academic Editor

PLOS ONE

Journal Requirements:

Additional Editor Comments:

Please see reviewers comments. Some minor revisions needed

Reviewers' comments:

Reviewer's Responses to Questions

**Comments to the Author**

1. If the authors have adequately addressed your comments raised in a previous round of review and you feel that this manuscript is now acceptable for publication, you may indicate that here to bypass the “Comments to the Author” section, enter your conflict of interest statement in the “Confidential to Editor” section, and submit your "Accept" recommendation.

Reviewer #3: All comments have been addressed

2. Is the manuscript technically sound, and do the data support the conclusions?

Reviewer #3: Yes

3. Has the statistical analysis been performed appropriately and rigorously? 

Reviewer #3: N/A

4. Have the authors made all data underlying the findings in their manuscript fully available?

Reviewer #3: Yes

5. Is the manuscript presented in an intelligible fashion and written in standard English?

Reviewer #3: Yes

6. Review Comments to the Author

Reviewer #3: Lines 55-57: data refer to yearly number of deaths. “yearly” should be added.

By 2050, if no action is taken against AMR, it is expected to cause 10 million deaths YEARLY worldwide, which exceeds the number of deaths caused by cancer [4, 5]

Lines 94-100: the same sentence is repeated twice!

Line 159: L.pneumophila instead of L. pneumoniae

Line 253: and roxithromycin was MOST COMMONLY prescribed for 1 disease

Line 258: Should this not be “2016-2918” instead of “2018”?

Line 269: Table 3. The word “Infectious” should be removed.

7. PLOS authors have the option to publish the peer review history of their article (what does this mean?). If published, this will include your full peer review and any attached files.

Reviewer #3: No

---

## [Author Response · Author response to Decision Letter 1]

13 May 2023

Please see the attached file "Response to Reviewers".

---

## [Decision Letter · Decision Letter 2]

4 Jun 2023

Investigation of oral macrolide prescriptions in Japan using a retrospective claims database, 2013–2018

PONE-D-22-29639R2

Dear Dr. Ide,

We’re pleased to inform you that your manuscript has been judged scientifically suitable for publication and will be formally accepted for publication once it meets all outstanding technical requirements.

Kind regards,

Iddya Karunasagar

Academic Editor

PLOS ONE

Additional Editor Comments (optional):

All reviewer comments have been addressed.

Reviewers' comments:

Reviewer's Responses to Questions

**Comments to the Author**

1. If the authors have adequately addressed your comments raised in a previous round of review and you feel that this manuscript is now acceptable for publication, you may indicate that here to bypass the “Comments to the Author” section, enter your conflict of interest statement in the “Confidential to Editor” section, and submit your "Accept" recommendation.

Reviewer #3: All comments have been addressed

2. Is the manuscript technically sound, and do the data support the conclusions?

Reviewer #3: Yes

3. Has the statistical analysis been performed appropriately and rigorously? 

Reviewer #3: N/A

4. Have the authors made all data underlying the findings in their manuscript fully available?

Reviewer #3: Yes

5. Is the manuscript presented in an intelligible fashion and written in standard English?

Reviewer #3: Yes

6. Review Comments to the Author

Reviewer #3: (No Response)

7. PLOS authors have the option to publish the peer review history of their article (what does this mean?). If published, this will include your full peer review and any attached files.

Reviewer #3: No

---

## [Editor Report · Acceptance letter]

13 Jun 2023

PONE-D-22-29639R2 

Investigation of oral macrolide prescriptions in Japan using a retrospective claims database, 2013–2018 

Dear Dr. Ide:

I'm pleased to inform you that your manuscript has been deemed suitable for publication in PLOS ONE. Congratulations! Your manuscript is now with our production department. 

Kind regards, 

on behalf of

Dr. Iddya Karunasagar 

Academic Editor

PLOS ONE